# Impact of the Timing and Use of an Insecticide on Arthropods in Cover-Crop-Corn Systems

**DOI:** 10.3390/insects13040348

**Published:** 2022-03-31

**Authors:** Gabriela Inveninato Carmona, Emily Robinson, Julia Nogueira Duarte Campos, Anthony Justin McMechan

**Affiliations:** 1Department of Entomology, University of Nebraska-Lincoln, Lincoln, NE 68588, USA; justin.mcmechan@unl.edu; 2Department of Statistics, University of Nebraska-Lincoln, Lincoln, NE 68588, USA; emily.robinson@huskers.unl.edu; 3Department of Entomology, University of Florida, Gainesville, FL 32611, USA; julia.nog.campos@gmail.com

**Keywords:** cover crop, insecticide, arthropods, corn

## Abstract

**Simple Summary:**

Cover crop use is increasing in the USA as a sustainable method. However, cover-crop pests can migrate to the following cash crop, threatening its productivity. As a preventative strategy to minimize pest transitions, growers may apply insecticides at the cover-crop termination time. Our study aims to better understand the impact of insecticide application as a preventive strategy against arthropods, either at cover-crop termination or when the cover crop is decomposing. Our finding indicates that preventive insecticide applications are not needed, highlighting the importance of scouting for pests before making a management decision. Moreover, we hypothesize that cover-crop biomass might create a physical barrier protecting arthropods below the cover-crop canopy.

**Abstract:**

Cover crops provide a habitat for pests and beneficial arthropods. Unexpected pest pressure in a cover-crop-to-corn system can occur and result in increased use of insecticides. Eight site-years of on-farm field studies were conducted in 2019, 2020, and 2021. The objective of the study was to evaluate the impact of insecticide timing relative to cover-crop termination on arthropod activity in a cover-crop-to-corn system. The treatments consisted of (i) glyphosate to terminate the cover crop, (ii) glyphosate and pyrethroid tank mix to terminate the cover crop, and (iii) glyphosate to terminate the cover crop and pyrethroid application 25 days after the termination. Arthropod activity was measured with pitfall traps before and at each treatment application. A total of 33,316 arthropods were collected. Total arthropods, Collembola, and Aphididae were the only taxa reduced with an insecticide application. The other arthropod taxa were mainly influenced by the sampling period. No significant pest pressure occurred at any site-year. Insecticide applications are not generally needed in a cover-crop-to-corn system. Scouting for pests and applying strategies only when necessary is crucial to conserve potentially beneficial arthropods in the system.

## 1. Introduction

The adoption of cover crops has increased from 10.3 million acres in 2012 to 15.4 million acres in 2017 [1]. In the USA, projections estimate that approximately 20 million acres were planted with cover crops in 2021, with a potential increase to 100 million acres by 2025 [2]. A nationwide survey performed by the North Central Region Sustainable Agriculture Research and Education [3], along with other studies, reported several well-documented sustainable benefits of using cover crops, such as improving herbicide-resistant weed control, an increase in soil quality, and an increase in cash crop yields [4,5,6,7,8]. Cereal rye (*Secale cereal* L.) is the most frequently used cover crop in the Midwest in soybean–corn rotations due to its ability to survive the winter and its greater biomass accumulation potential in the spring [9]. In such systems, cereal rye can be seeded after soybean or corn after harvest, where it begins to grow during the fall and over winter, and then resumes growth in the spring [10,11].

Due to the sustainable benefits of using cover crops, national and state programs have provided financial assistance to growers to encourage the adoption of cover crops. The Environmental Quality Incentives Program (EQIP) and the Conservation Stewardship Program (CSP) are the most well-known of these programs. EQIP payments vary by state, ranging from $62.33 (Illinois) to $92.27 (Delaware), where almost 2.5 million cover crop acres were enrolled in the program in 2018 [1]. The cover-cropped acres estimated to be part of the CSP program increased from 350,000 acres in 2010 to more than 2 million acres in 2015 [1]. Such national and state initiatives to support cover-crop adoption highlights efforts to increase cover-crop use in the USA. However, the best management strategies to maximize cover-crop benefits and decrease the risk of pests in this system are not well understood.

Polyphagous cover-crop pests have the potential to transition to the following cash crop and become a threat to cash crop yield. A recent study mentioned several moths as being a risk in cereal rye–corn cropping systems, such as true armyworm (*Mythimna unipuncta*; Lepidoptera: Noctuidae), black cutworm (*Agrotis ipsilon* Hufnagel; Lepidoptera: Noctuidae), and common stalk borer (*Papaipema nebris* Guinee; Lepidoptera: Noctuidae) [12]. The authors mention that transition risk increases as those pests feed on the cereal rye cover crop and are early-season corn pests. The transition time of the cover-crop pest to the cash crop is expected to occur when the cover crop is deteriorating. However, the transition timing of the pest also depends on its biology and feeding habit.

To maximize cover-crop benefits, growers tend to delay termination to increase cover-crop biomass production. The CTIC survey reported that 54.3% of the 1172 surveyed farmers in the USA planted their cash crop into a live cover crop, and 68% of those reported better soil moisture management. However, cover-crop termination timing relative to the cash crop planting is a concern with regards to insect pest transitions from cover crops to the following cash crop [3]. Recently, the increased likelihood of wheat-stem maggots (*Meromyza americana* Fitch; Diptera: Chloripidae) transitioning from cereal rye or wheat cover crop to corn when the cover crop was terminated after planting corn resulted in corn yield losses of up to 30 bushels per acre (2017 kg per hectare) [13]. In addition, the same CTIC survey reported that farmers planting cash crops into a living cover crop had the potential to increase cutworm issues in fields. Due to increased risk in insect pest transitioning from the cover crop to the following cash crop and low product cost, farmers have been known to apply pyrethroid insecticides as a tank mix with an herbicide at the cover-crop termination time in an attempt to minimize the risk of a pest transition from the cover crop to the following cash crop.

Currently, the primary insect pest management strategy adopted by growers in a conventional system is the use of insecticides [14]. Pyrethroid insecticides are often used due to their fast action, easy application, low cost, and potential economic return. However, unneeded or preventive applications often occur on a farm scale, negatively impacting arthropod conservation and biodiversity, increasing insecticide resistance, and consequently reducing insecticide efficacy against target pests [15]. Scouting agricultural systems to identify pest levels and justify management is a fundamental strategy in any cropping system [16]. However, there is no information about whether insecticide applications at the cover-crop termination time are efficient, sustainable, or even profitable. Research is needed to address the impact of insecticide management of arthropods in cover-crop-cash crop rotation systems to help guide the use of better management practices. As a result, on-farm field studies were conducted across eight site-years in eastern Nebraska to evaluate (1) insecticide timing of application impact on arthropod activity in the following corn and (2) identify the best arthropod-management strategy to assist farmers in making profitable and sustainable decisions. We hypothesized that if cover-crop pests are present, such as wheat stem maggots, those pests would be reduced with late insecticide application. We hypothesized that the addition of any preventive insecticide application would reduce arthropod activity and would not increase corn yields unless significant pest pressure occurred.

## 2. Materials and Methods

### 2.1. Experimental Design and Field Characteristics

Eight site-years of on-farm field studies (hereafter: sites) were conducted on rainfed growers’ fields in eastern Nebraska, three in 2018/19, three in 2019/20, and two in 2020/21 (Table 1). The experiment was conducted as a randomized complete-block design with four replications with the exception of site seven, where a Latin-Square design was used due to the combination of field slope and field edge. Treatments consisted of (i) glyphosate to terminate the cover crop (hereafter: herbicide only), (ii) glyphosate and pyrethroid at cover-crop termination (hereafter: tank-mix), and (iii) glyphosate to terminate the cover crop and a pyrethroid application 25 days after the cover-crop termination (hereafter: late insecticide). Late insecticide treatment was selected, as it is a potentially critical time for pest movement to the cash crop. Cereal rye was planted by the growers in the previous fall and terminated after planting corn. The corn planting dates varied between sites, ranging from three to twelve days after terminating the cover crop (Table 2). Experiments conducted in 2018/19 had smaller plot sizes than the experiments conducted in 2019/20 and 2020/21. The plot size varied between sites according to the area available (Table 2). In 2018/19 and 2019/20, 142 g·ha^−1^ of pyrethroid insecticide (HERO^®^; a.i: zeta-cypermethrin and bifenthrin) with ammonium sulfate was added to the carrier water (90.3 L per hectare) as the tank-mix and late insecticide application was used. In 2020/21, 283 g·ha^−1^ of pyrethroid insecticide (HERO^®^) with the same carrier rate. Cover crop growth/status on the day of the late insecticide application is shown in Figure A1. The cover crop was terminated using an herbicide, Roundup PowerMAX, a.i: glyphosate (*N*-(phosphonomethyl) glycine^3^) (2.24 kg·ha^−1^). Crop management and treatment application dates for each site and year are described in Table 1 and Table 2, respectively.

### 2.2. Arthropod Sampling

Pitfall traps were used to capture ground-dwelling arthropod activity. One pitfall trap was placed in the center of each plot. A circular pitfall trap was used. Each pitfall trap consisted of a 473 mL cup sunken into the ground with the rim level with the soil surface. A removable 236 mL collecting cup was placed inside the larger cup, and 170 mL of propylene glycol-based antifreeze liquid with no attractant was added for each collection period to immobilize arthropods for further identification. During the pitfall collection period, a thick plastic plate was used as a cover to limit the impact of rain, at a height of 5 cm from the soil surface. A total of three pitfall samples were taken during each growing season from each plot, 15 days before any treatment application (hereafter: sample one), at the cover-crop termination/tank mix (hereafter: sample two), and the late insecticide treatment applications (hereafter: sample three). The pitfall traps were active in the field for an average of five days with the specific dates per site shown in Table 2. The content of the pitfall traps was transferred to an individual 354 mL labeled whirl bag for further analysis. All insects were counted and identified to the family level, while all other arthropods collected were identified to the order level.

### 2.3. Corn Injury Assessment

The corn plants were evaluated for signs of insect presence and feeding injury above and below ground during the V3 corn development stage. Two 5 m rows were randomly selected in one location from each plot to be evaluated for un-emerged or under-developed plants. Additional observation notes were made if needed. When insects were found causing injury to corn plants, they were collected and placed in sealed plastic bags for further identification.

### 2.4. Agronomic Parameters

The cover-crop extended leaf height and biomass were measured at the cover-crop termination date for each site. The cover-crop extended leaf height was measured from the soil surface to the tallest extended leaf at three locations in each plot and was randomly chosen by walking diagonally across each plot. The cover-crop biomass was sampled in two locations on each plot, covering a total area of 0.38 m^2^ per experimental unit. Samples were collected by cutting cover crops and weeds just above the soil surface within a PVC rectangle area of 0.19 m^2^. Weeds, when present, were placed in a separate bag for each plot. All samples were placed in a dryer at 75 °C until (72 h) a constant weight was reached. Dry weights were recorded, and cover-crop biomass per hectare per treatment was calculated. At the end of the season, 5 m from the middle two rows of each plot were hand-harvested, and the corn grain yield per hectare was recorded.

### 2.5. Statistical Analysis

#### 2.5.1. Arthropod Activity

All pitfall data was standardized as the number of arthropods collected per 96 h period to avoid bias based on the length of the sampling periods. Total arthropods and arthropod taxa that corresponded to more than 1% of the total arthropods capture were individually analyzed using analysis of covariance (ANCOVA) and Generalized Linear Mixed Model (PROC GLIMMIX) following a negative binomial distribution with a log function in SAS (SAS Institute, version 9.4) [17]. For the total number of arthropods and individual arthropod taxa, a baseline count (covariate) was used as a covariate in the ANCOVA model. The baseline count was obtained from the sampling period before any treatment application to account for the initial differences in arthropod populations across each site-year. Each site-year was analyzed separately due to confounding variables, such as crop management and treatment application dates. The treatment, sample period, and the treatment by sample period interaction were classified as fixed effects with the baseline count as a covariate. Random effects were rep and treatment nested in rep. Estimated treatment means were calculated at the average baseline value for each arthropod taxa per site. In site three, the pitfall before treatment applications was lost due to heavy rain, so we were not able to perform ANCOVA analysis for this site. For each site, the analysis uses pairwise comparison tests to control for Type I error rates. Tukey LSD is reported at an α=0.05 significance level. The SLICEDIFF statement in SAS was used to test the pairwise differences between treatments when the sample period main effect was significant. Corn injury assessments were not analyzed, because of low pest pressure (>1%) in all sites and years.

#### 2.5.2. Agronomic Parameters

A Linear Mixed Model (PROC GLIMMIX) was run in SAS 9.4 with a two-way ANOVA treatment design to determine the effect of treatment and site on cover-crop biomass, cover crop extended leaf height, and corn grain yield. Agronomic parameters were analyzed using treatments and sites as fixed effects, while rep and treatment nested in rep were considered random effects. A Tukey adjustment was used on pairwise-comparison tests to control for Type I error rates. Tukey LSD is reported at α=0.05 significance level.

## 3. Results

### 3.1. Arthropod Activity

The pitfall traps collected 33,316 total arthropods across all sites and years of the study. During the 2019 growing season (sites one, two, and three), a total of 13,292 individual arthropods were collected representing 42 different taxa. For the 2020 growing season (sites four, five, and six), a total of 14,028 individual arthropods were collected, representing 38 different taxa. In 2021 (sites seven and eight), a total of 6026 individual arthropods were collected, representing 33 different taxa.

#### 3.1.1. 2019 Growing Season

For 2019, the most dominant taxa were Collembola (39.9%), Acari (35.6%), Coleoptera: Zopheridae (4.8%), Hemiptera: Aphididae (4.5%), Coleoptera: Nitidulidae (2.0%), Coleoptera: Staphylinidae (1.9%), Diptera: Anthomyiidae (1.8%), Coleoptera: larvae (1.7%), Araneae (1.6%), and Coleoptera: Carabidae (1.5%), representing 95.3% of total arthropods collected (Table 3).

Total arthropod and Aphididae activity from site three were the only taxa impacted by the study treatments. For total arthropod activity, a significant interaction occurred between treatment and sampling period (Table 4; Figure 1A). A multiple-treatment test within the sample dates indicated that the interaction was due to a lack of differences between the treatment means during sample two (*F* = 0.22; *df* = 2, 9; *p* = 0.8056), while the herbicide only (387) had greater total arthropod activity compared to the tank mix (159; *t* = 3.87; *df* = 9; *p* = 0.0096) and late insecticide (136; *t* = 4.02; *df* = 9; *p* = 0.0077) treatments in sample period three. The treatment effect was approaching significance for Aphididae as a result of greater arthropod activity in the herbicide only (15), followed by the late insecticide (4), and then the tank mix (0.1) treatments (Table 4; Figure 1B).

#### 3.1.2. 2020 Growing Season

For 2020, the most abundant taxa were Collembola (27.5%), Coleoptera: Zopheridae (21.3%), Coleoptera: Nitidulidae (16.7%), Hemiptera: Aphididae (15.2%), Acari (3.9%), Araneae (3.3%), Diptera: Anthomyiidae (3.0%), Coleoptera: larvae (2.7%), and Coleoptera: Carabidae (1.7%), representing 95.1% of the total arthropod collected (Table 3).

In 2020, Aphididae and Carabidae activity in site five were the only taxa impacted by the study treatments. A significant interaction between the treatment and sampling period was identified for Aphididae (Table 4; Figure 1C) and Carabidae (Table 4; Figure 1D). For Aphididae, multiple treatment tests within sample dates indicated that the interaction was a result of a lower Aphididae activity in the tank mix treatment (17) compared to the herbicide only (72; *t* = 5.56; *df* = 9; *p* = 0.0004) and late insecticide (68; *t* = 5.40; *df* = 9; *p* = 0.0011) treatments during sample two. In contrast, no differences between treatment means were observed during sample three (*F* = 0.64; *df* = 2, 9; *p* = 0.5477) (Table 4; Figure 1C). The interaction between the treatments and the sampling period for Carabidae activity was the result of greater Carabidae activity in the tank mix treatment (7) compared to the herbicide only (0.08; *t* = −3.14; *df* = 9; *p* = 0.0291) and late insecticide (0.05; *t* = −3.29; *df* = 9; *p* = 0.0230) in sample two, while no differences between treatments occurred in sample three (*F* = 0.40; *df* = 2, 9; *p* = 0.6834) (Table 4; Figure 1D).

#### 3.1.3. 2021 Growing Season

For the 2021 growing season, the most abundant taxa were Collembola (54.3%), Acari (17.1%), Coleoptera: Zopheridae (6.0%), Diptera: Sciaridae (5.6%), Coleoptera: Nitidulidae (5.2%), Araneae (3.4%), Chilopoda (1.7%), and Coleoptera: Carabidae (1.5%), representing 95.1% of the total arthropods collected (Table 3).

The total arthropod and Collembola activity in site eight were the only taxa influenced by the study treatments. The interaction between treatment and sampling period was significant for the total arthropod (Table 4; Figure 1E) and Collembola activity (Table 4; Figure 1F). For total arthropod activity, a multiple treatment test within sample dates found a similar response in treatments for sample two (*F* = 2.91; *df* = 2, 11; *p* = 0.1306). In contrast, the total arthropod activity was greater for the herbicide-only (234) treatment compared to the tank mix (121; *t* = 3.59; *df* = 6; *p* = 0.0267) and late insecticide (91; *t* = 2.96; *df* = 6; *p* = 0.0254) for sample three (Table 4; Figure 1E), resulting in an interaction between treatments for the two sample dates. For Collembola, a multiple treatment test within sample dates indicated that the interaction was a result of greater activity in the tank mix treatment (126) compared to the herbicide only (59; *t* = −3.65; *df* = 6; *p* = 0.0107) and late insecticide (43; *t* = −2.18; *df* = 6; *p* = 0.0425) for sample two. In sample three, the herbicide-only treatment (31) had greater Collembola activity compared to the tank mix (14; *t* = 2.41; *df* = 6; *p* = 0.0325) and to the late insecticide (11; *t* = 2.53; *df* = 6; *p* = 0.0419) treatments (Table 4; Figure 1F). Although Collembola was the most abundant arthropod overall, Acari was the most abundant during the late insecticide application sample period driving the interaction at that sample period. The results from the other arthropod taxa that composed more than 1% of the total arthropod activity were only affected by the sampling period Table 4.

### 3.2. Agronomic Parameters

#### 3.2.1. Cover-Crop Extended Leaf Height

All sites in this study met the cover-crop extended leaf height NRCS threshold (152 mm) for cover-crop termination at the termination time (Figure 2A). The extended leaf height varied between sites (*F* = 156.91, *df* = 7, 69, *p* = < 0.0001) with sites seven and eight having the greatest extended leaf-height mean, followed by site six, then site two. Sites three, four, and five did not differ from each other but had lower extended leaf heights compared to sites two, six, seven, and eight. Site one had a significantly lower extended leaf-height mean compared to all other sites (Figure 2A).

#### 3.2.2. Cover-Crop Biomass

The cover-crop biomass varied from 629 kg·ha^−1^ (site three) to 4806 kg·ha^−1^ (site eight) (Figure 2B). The cover-crop biomass varied between sites (*F* = 70.30, *df* = 7, 69, *p* < 0.0001). Site eight had the greatest cover-crop biomass (4806 kg·ha^−1^), followed by sites seven (3946 kg·ha^−1^) and six (3744 kg·ha^−1^), and then site one (2013 kg·ha^−1^). Site one was not different from sites two (1897 kg·ha^−1^) and four (1927 kg·ha^−1^), but it was greater than sites five (1019 kg·ha^−1^), and three (629 kg·ha^−1^). Finally, sites two and four had greater cover-crop biomass compared to site three but did not differ from site five (Figure 2B).

#### 3.2.3. Corn Grain Yield

The corn yield varied from 7831 (site one) to 13,208 kg·ha^−1^ (site seven). Significant differences in corn-grain yield occurred between sites (*F* = 13.32; *df* = 7, 89; *p* < 0.0001), while treatment effect was not significant (*F* = 0.29; *df* = 2, 89; *p* = 0.7496). Site seven had the greatest corn grain yield mean (13,208 kg·ha^−1^), followed by site five (12,003 kg·ha^−1^), eight (10,540 kg·ha^−1^), four (10,011 kg·ha^−1^), two (9998 kg·ha^−1^), three (9866 kg·ha^−1^), six (9856 kg·ha^−1^), and the lowest yield was recorded in site one (7831 kg·ha^−1^) (Figure 3).

### 3.3. Environmental Conditions during the Pitfall Trap Sample Periods

The cumulative average temperature and precipitation varied between sites and sampling periods (Table A1). During 2019, the cumulative average temperature varied from 127.5 °C (sample one, site one) to 210.6 °C (sample two; site one). Cumulative precipitation varied from 1.3 (samples one and two; site two) to 43.9 mm (sample two; site one). For 2020, the cumulative average temperature varied from 125.3 °C (sample one; sites five and six) to 200.0 °C (sample three; site four). The cumulative precipitation varied from 0.0 (sample two; site four, samples one and two for sites five and six) to 10.9 mm (sample three; site six). Finally, during 2021, the cumulative average temperature varied from 79.4 °C (samples one and two for sites seven and eight) to 194.4 °C (sample three; site eight). The cumulative precipitation varied from 0.0 (sample one; sites seven and eight, sample three; site eight) to 7.9 mm (sample three; site seven).

## 4. Discussion

This is the first multi-site-year on-farm study, to our knowledge, to evaluate insecticide management in a cover-crop-corn rotation system. Cover-crop use is rapidly increasing, and growers need scientific information to guide sustainable and profitable management decisions [2]. Our study showed that insecticide application as the tank mix could reduce Aphididae activity, an insect group present in the upper canopy of the cover crop, and an increase in Carabidae activity were observed in the same treatment at the same site. Based on previous similar findings, we believe that the Carabidae activity increase was led by a prey–predator relationship with dead Aphididae in the same plots [18,19,20,21,22,23]. However, most of the arthropods evaluated were not affected by any insecticide application. Those findings led to the hypothesis that the cover-crop biomass created a physical shelter–barrier that may have protected beneficial and potential insect pests at the lower cover-crop canopy from the contact insecticide application; however, further research to test this hypothesis is needed. The reduction in total arthropod activity only with late insecticide application provides insights supporting this hypothesis, as the cover crop was decomposing, and the cover crop’s physical shelter–barrier was reduced when the late insecticide application was made.

Increases in ground beetles (Coleoptera: Carabidae) captured by pitfall traps after insecticide applications appear to be surprisingly common [18,19,20,21,22,23]. Indirect and direct effects have been cited in the literature as possible causes for this increase. An increase in collected carabids after 17 days of pyrethroid application was reported in the literature [19]. The authors suggested that the reason for it was carabid movement into the insecticide-treated plots to feed on dead and impaired insects. Indirect mechanisms such as changes in mobility, abundance, and distribution of prey have also been reported as the reason for an increase in Carabidae activity-density after insecticide application [19,21]. In addition, low doses of insecticide might trigger a hormesis effect on arthropods. Hormesis is characterized by a low-dose response that is opposite in effect to that seen at high doses Moreover, it has been reported that hormesis responses can accelerate insect population growth [22,23,24]. Our results indicated increases in Carabidae activity after tank-mix application. The results of our study also showed that Aphididae was reduced in the same treatment and that the same site that Carabidae activity increased with the tank-mix treatment. We hypothesize that the increase in Carabidae activity was led by a prey–predator relationship with dead Aphididae at the same treatments or by a hormesis effect.

Early pest pressure in the cover-crop-corn system has been reported, such as true armyworm, black cutworm, and wheat-stem maggot [12,13]. However, no significant pest pressure was observed in any site-year of this study. The lack of pest pressure might be explained by growers using Bt hybrids, which control most of the early-season corn pests, or due to natural low pest pressure in the area. However, Bt hybrids are not effective against wheat-stem maggot. A recent experiment in a cover-crop-corn system conducted in eastern and central Nebraska, using either Bt or non-Bt corn hybrids, reported less than 1% of pest pressure in their studies [25].

Pest transition from the cover crop to corn has been recently reported [12,13]. Field surveys of wheat-stem maggot injury to corn from cover crops suggested that terminating the cover crop before planting corn or having the cover crop completely dead before the cash-crop planting is a strategy to minimize pest transitions [13]. However, maximizing cover-crop biomass is often a primary goal for growers to increase cover-crop benefits, such as weed suppression, erosion control, and water quality improvement [1,4,8]. As a result, growers are hesitant to terminate the cover crop too early. The occurrence of pest pressure in this system can lead some growers to add pyrethroid insecticides, often applied with other chemicals as a tank mix at the cover-crop termination time to reduce any possible pest transition to the following cash crop. In our study, we hypothesized that if a pest such as wheat-stem maggot were present, pest pressure could be reduced if a late insecticide application coincided with larval movement between the cover crop to corn. Nonetheless, no pest pressure was present in any site-year, so the efficacy of insecticide management against pests in this system could not be evaluated. The lack of pest pressure in this study highlights the fact that pests are infrequent in a cereal rye to corn system, and the use of an insecticide is often not warranted.

Insecticide applications as a preventive strategy are frequently used by growers, especially when pest populations in the area have been a problem [26]. Unnecessary insecticide application might decrease total arthropod activity in the system. This reduction might impact the arthropods’ prey–predato dynamics, potentially making beneficial insects less effective in controlling pest populations or potentially resulting in secondary pest outbreaks due to an imbalanced system [27,28,29,30,31,32,33]. We hypothesized that any preventive insecticide application in a cover-crop system would reduce the total arthropod activity. The results of our study partially supported our hypothesis as insecticide applications reduced the total number of arthropods in two site years. Even though carabids, a beneficial arthropod, increased with the tank-mix application, it might be a disadvantage for the sustainability of the system. We believe that the increased activity of Carabidae was the result of abundant dead or immobilized prey; therefore, Carabids could potentially be exposed to pesticides as well. Additional studies will be needed to evaluate any potential sublethal effects from the consumption of prey exposed to a pyrethroid in a cover-crop system. If sublethal effects occur, it could make Carabids less effective in controlling pests. The observed reduction in total arthropods and the potential disadvantage of Carabid exposure to a pyrethroid reinforces the need to avoid unnecessary insecticide applications in cover crops when pest pressure is low.

Despite the decrease or increase in activity of some arthropod taxa in our study, most of the taxa evaluated were not impacted by any of the insecticide applications. We hypothesize that the lack of insecticide impact on arthropods in our on-farm studies could result from cover-crop biomass sheltering the soil-dwelling arthropods against the insecticide applications. However, this study was not designed to address the role of biomass protection for ground-dwelling arthropods. A pyrethroid is a broad-spectrum, contact insecticide. Therefore, barriers that protect the ground-level arthropods from contact insecticides, such as cover-crop biomass accumulation, might limit the insecticide from reaching arthropods below the cover-crop canopy. Moreover, this could pose an issue for insecticide efficacy against insect pests present in the cover-crop lower canopy. In addition, the reduction in Aphididae activity was expected as Aphididae is often on the cover-crop canopy, and, as a result, they are vulnerable to insecticide contact applications. In addition, the Aphididae species found in this study do not pose any significant threat to vegetative-stage corn.

## 5. Conclusions

This research assists growers in making informed, profitable, and sustainable decisions that could result in a decrease in unneeded insecticide use. Insecticide applications did not impact corn-grain yields as was hypothesized. With no pest pressure at any site in the study, insecticide applications were unnecessary, resulting in additional expense, labor, and time for growers with no return on investment. Finally, the infrequent presence of pests in a cover-crop-to-corn system highlights the importance of scouting for pests prior to any management decision. Future research will be needed to address the role of cover-crop biomass production and insecticide management to test the hypothesis that cover-crop biomass production creates a physical shelter–barrier, protecting ground-dwelling arthropods. More information regarding the side effects on Carabidae when feeding on pesticide-exposed prey will also be important to evaluate. Future research should consider artificial pest infestations to further evaluate the impact of insecticide use in cover-crop systems and corn-grain yield trade-offs. In addition, predator gut analyses should be performed in future studies to better understand the predator–prey relationships in a cover-crop-corn cropping system.

## Figures and Tables

**Figure 1 insects-13-00348-f001:**
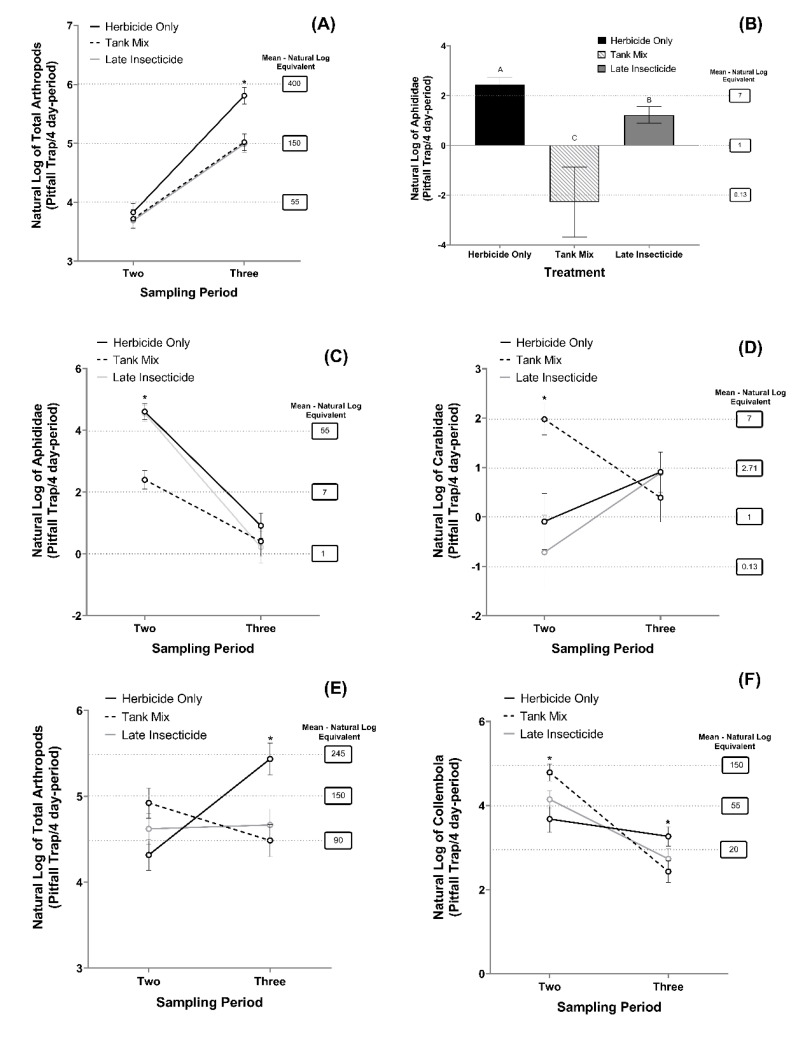
Natural log of total arthropod activity per sampling period and treatment from site 3 (**A**), Aphididae activity per treatment from site 3 (**B**), Aphididae activity per sampling period and treatment from site 5 (**C**), Carabidae activity per sampling period and treatment from site 5 (**D**), of total arthropod activity per sampling period and treatment from site 8 (**E**), and of Collembola activity in log per sampling period and treatment from site 8. Sampling period two was performed at the tank-mix application and three at the late insecticide application time (**F**). Error bars indicate the standard error of the natural log of the means. * Represents statistical difference at *p* < 0.05 between treatments at a given sampling period. Same letters represent no statistically significant difference at *p* < 0.05. Grey dash lines indicate the natural log–mean equivalent.

**Figure 2 insects-13-00348-f002:**
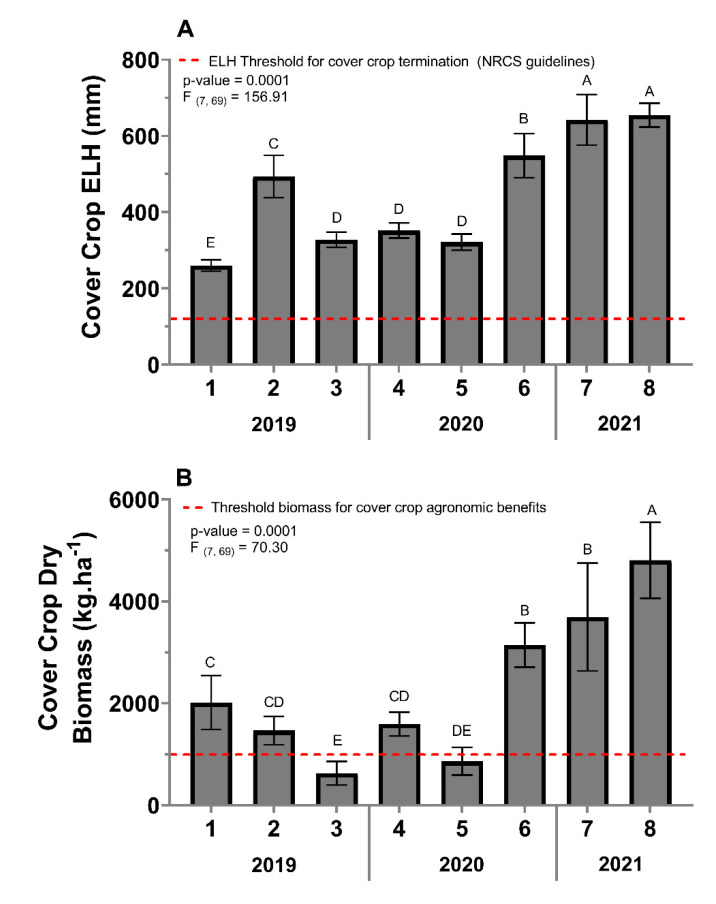
Cover crop extended leaf height (ELH) (**A**) and cover-crop biomass (**B**) per site taken at cover-crop termination. Error bars indicate the standard error of the means. The same letters represent no statistically significant difference at *p* < 0.05.

**Figure 3 insects-13-00348-f003:**
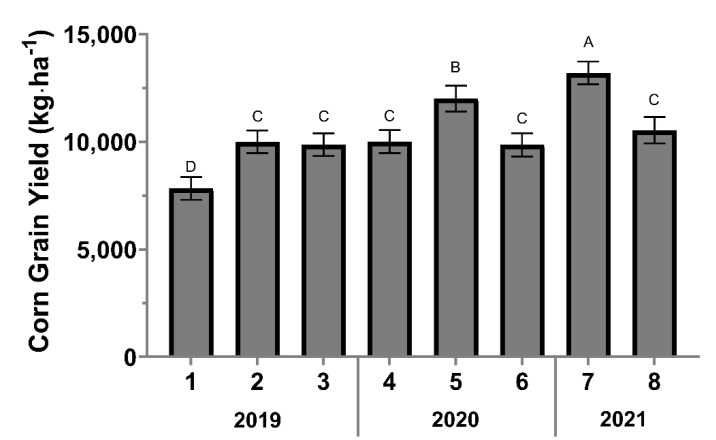
Corn grain yield in kg·ha^−1^ per site. Error bars indicate the standard error of the means. The same letters represent no statistically significant difference at *p* < 0.05.

**Table 1 insects-13-00348-t001:** Cereal rye and corn management dates and specifications per site and year.

Year	Site	Nebraska County	Cover Crop Management			Corn Management			
			Cover Crop Planting Dates	Seed Rate (kg·ha^−1^)	Row Spacing (cm)	Corn Planting	Seed Rate (seeds·ha^−1^)	Corn Hybrid	Corn Harvest
2019	1	Saunders	24 October 2018	106	19	28 April 2019	79,040	P1197AM	24 October 2019
	2	Saunders	Late November, 2018	67	19	28 April 2019	74,131	DKC60-88	20 October 2019
	3	Lancaster	Late November, 2018	73	38	15 April 2019	86,486	DKC63-90 RIB	9 October 2019
2020	4	Lancaster	Late October, 2019	67	19	23 April 2020	74,131	DKC60-88	15 October 2020
	5	Saunders	Late October, 2019	73	Fly	30 April 2020	79,040	P1366AM	11 October 2020
	6	Saunders	Late September, 2019	67	19	28 April 2020	86,486	DKC63-90 RIB	24 October 2020
2021	7	Saunders	Late October, 2020	67	38	26 April 2021	79,040	P1366AM	4 October 2021
	8	Lancaster	Late October, 2020	78	Fly	1 May 2021	86,486	DKC63-90 RIB	4 October 2021

**Table 2 insects-13-00348-t002:** Treatment application dates, details, and measurement information per site and year.

Year	Site	Nebraska County	Treatment Applications				Measurements				
			Cover-Crop termination and Tank Mix Application	Late Insecticide Application	Insecticide Rate (g·ha^−1^)	Plot Size (m × m)	Sampling Periods (Pitfall Traps)			Corn Injury Assessment	Cover-Crop biomass
							I	II	III		
2019	1	Saunders	6 May 2019	22 May 2019	142	9.14 × 9.14	24–28 April 2019	6–13 May 2019	Lost	30 May 2019	6 May 2019
	2	Saunders	2 May 2019	1 June 2019	142	12.16 × 12.16	20–24 April 2019	2–6 May 2019	13–17 May 2019	30 May 2019	2 May 2019
	3	Lancaster	2 April 2019	3 June 2019	142	9.14 × 9.14	1–5 May 2019	22–17 May 2019	3–7 June 2019	10 June 2019	22 May 2019
2020	4	Lancaster	27 April 2020	11 May 2020	142	27.4 × 30.48	30 March–6 April 2020	27 April–2 May 2020	11–17 May 2020	29 May 2020	27 April 2020
	5	Saunders	1 May 2020	18 May 2020	142	27.4 × 30.48	6–10 April 2020	1–6 May 2020	18–23 May 2020	4 June 2020	1 May 2020
	6	Saunders	1 May 2020	18 May 2020	142	27.43 × 27.43	6–10 April 2020	1–6 May 2020	18–23 May 2020	4 June 2020	1 May 2020
2021	7	Saunders	30 April 2021	22 May 2021	283	27.43 × 27.43	12–15 April 2021	29 April–4 May 2021	22–26 May 2021	1 June 2021	30 April 2021
	8	Lancaster	13 May 2021	3 June 2021	283	27.4 x 30.48	12–15 April 2021	12–17 May 2021	3–7 June 2021	7 June 2021	13 May 2021

**Table 3 insects-13-00348-t003:** Arthropod taxa activity, the percentage from total activity, and the classification of arthropod taxa that composes more than 1% of the total arthropod activity per site and year.

Year	Class		Insecta								Arachnida		Chilopoda	Collembola	Total Arthropods
	Order		Hemiptera	Diptera		Coleoptera									
	Family		Aphididae	Anthomyiidae	Sciaridae	Zopheridae	Staphylinidae	Nitidulidae	Carabidae	Larvae	Araneae	Acari			
2019	Site 1	Number	24	43	67	58	86	64	36	50	68	1629	7	574	2880
		%	0.8	1.5	2.3	2	3	2.2	1.3	1.7	2.4	56.6	0.2	19.9	100
	Site 2	Number	0	145	35	79	135	27	40	39	43	2130	13	4442	7245
		%	0	2	0.5	1.1	1.9	0.4	0.6	0.5	0.6	29.4	0.2	61.3	100
	Site 3	Number	571	50	2	504	30	174	127	131	104	957	0	281	3137
		%	18.2	1.6	0.1	16.1	1	5.5	4	4.2	3.3	30.5	0	9	100
2020	Site 4	Number	1139	0	20	1463	39	451	145	31	211	220	6	1067	5029
		%	22.6	0	0.4	29.1	0.8	9	2.9	0.6	4.2	4.4	0.1	21.2	100
	Site 5	Number	843	316	22	1191	37	941	65	16	133	274	83	1870	5875
		%	14.3	5.4	0.4	20.3	0.6	16	1.1	0.3	2.3	4.7	1.4	31.8	100
	Site 6	Number	154	104	25	345	35	931	26	326	119	48	39	914	3124
		%	4.9	3.3	0.8	11	1.1	29.8	0.8	10.4	3.8	1.5	1.2	29.3	100
2021	Site 7	Number	3	24	95	255	5	107	69	7	135	154	101	1869	2870
		%	0.1	0.8	3.3	8.9	0.2	3.7	2.4	0.2	4.7	5.4	3.5	65.1	100
	Site 8	Number	41	4	266	108	13	206	21	0	71	874	0	1405	3157
		%	1.3	0.1	8.4	3.4	0.4	6.5	0.7	0	2.2	27.7	0	44.5	100

**Table 4 insects-13-00348-t004:** Analysis of covariance with significant interactions or main effects per site and year for each arthropod taxa composing more than 1% of the total arthropod.

Year	Class			Insecta								Arachnida		Collembola	Total Arthropods
	Order			Hemiptera	Diptera		Coleoptera					-		-	
	Family			Aphididae	Anthomyiidae	Sciaridae	Zopheridae	Staphalynidae	Nitidulidae	Carabidae	Larvae	Araneae	Acari	-	
2019	Site 1	SMP	F	n/a	10.78	n/a	12.59	16.68	3.58	8.19	4.91	1.11	85.41	20.17	66.98
			df ^2^		1, 13		1, 13	1, 13	1, 12	1, 13	1, 13	1, 13	1, 13	1, 13	1, 12
			P		**0.009**		**0.006**	**0.001**	0.083	**0.013**	0.095	0.310	**<0.0001**	**0.001**	**<0.0001**
		Trt	F		0.54		0.91	0.88	0.64	0.34	2.58	1.88	0.21	1.45	0.58
			df		2, 13		2, 13	2, 13	2, 12	2, 13	2, 13	2, 13	2, 13	2, 13	2, 13
			P		0.612		0.466	0.452	0.544	0.715	0.114	0.192	0.813	0.270	0.573
		SMP*Trt	F		0.09		0.16	1.2	0.12	0.66	0.81	0.07	0.01	3.2	0.67
			df		2, 13		2, 13	2, 13	2, 12	2, 13	2, 13	2, 13	2, 13	2, 13	2, 13
			P		0.979		0.856	0.333	0.885	0.533	0.464	0.928	0.991	0.074	0.527
	Site 2	SMP	F	n/a	0.87	n/a	8.11	7.61	5.75	2.75	5.42	n/a	0.36	4.84	5.43
			df		1, 9		1, 9	1, 9	1, 9	1, 9	1, 9		1, 9	1, 9	1, 9
			P		0.3763		**0.019**	**0.022**	**0.040**	0.132	**0.0450**		0.565	0.055	**0.045**
		Trt	F		4.4		1.17	1.59	0.54	0.11	3.57		2.46	1.74	3.87
			df		2, 9		2, 9	2, 9	2, 9	2, 9	2, 9		2, 9	2, 9	2, 9
			P		0.097		0.354	0.256	0.601	0.894	0.072		0.142	0.230	0.081
		SMP*Trt	F		5.08		0.23	2.06	0.2	0.3	0.81		1.21	0.05	0.41
			df		2, 9		2, 9	2, 9	2, 9	2, 9	2, 9		2, 9	2, 9	2, 9
			P		0.083		0.796	0.183	0.825	0.746	0.475		0.343	0.947	0.674
	Site 3 ^1^	SMP	F	2.75	15.85	n/a	0.83	4.69	18.65	51.02	22.95	25.14	116.07	95.92	247.11
			df	1, 9	2, 9		1, 9	1, 9	1, 9	1, 9	1, 9	1, 9	1, 9	1, 9	1, 9
			P	0.132	**0.003**		0.387	0.059	**0.002**	**<0.0001**	**0.001**	**0.001**	**<0.0001**	**<0.0001**	**<0.0001**
		Trt	F	4.19	0.61		1.09	1.17	1.68	0.52	2.93	0.87	2.13	0.67	4.52
			df	2, 9	2, 9		2, 9	2, 9	2, 9	2, 9	2, 9	2, 9	2, 9	2, 9	2, 9
			P	**0.052**	0.565		0.376	0.353	0.240	0.610	0.105	0.452	0.175	0.536	0.044
		SMP*Trt	F	2.08	0.28		1.41	0.73	1.28	0.4	2.27	0.56	0.22	0.91	5.57
			df	2, 9	2, 9		2, 9	2, 9	2, 9	2, 9	2, 9	2. 9	2, 9	2, 9	2, 9
			P	0.187	0.766		0.294	0.509	0.323	0.682	0.159	0.452	0.810	0.438	**0.027**
2020	Site 4	SMP	F	34.45	n/a	n/a	42.42	n/a	61.51	0.05	n/a	47.29	n/a	17.02	0.67
			df	1, 15			1, 12		1, 12	1, 12		1, 15		1, 12	1, 15
			P	<0.0001			**<0.0001**		**<0.0001**	0.0945		**<0.0001**		**0.002**	**0.001**
		Trt	F	2.3			0.63		0.94	0.64		0.96		2.31	0.24
			df	2, 15			2, 12		2, 12	2. 12		2, 15		2, 12	2, 15
			P	0.119			0.612		0.468	0.601		0.437		0.157	0.843
		SMP*Trt	F	1.3			0.26		1.24	2.8		0.17		0.42	0.13
			df	2, 15			2, 12		2, 12	2, 12		2, 15		2, 15	2, 15
			P	0.311			0.855		0.345	0.078		0.904		0.743	0.938
	Site 5	SMP	F	112.17	9.77	n/a	0.06	n/a	31.79	0.84	n/a	0.42	15.78	31.52	0.14
			df	1, 10	1, 10		1, 10		1, 10	1, 10		1, 10	1, 10	1, 10	1, 10
			P	**<0.0001**	**0.012**		0.817		**<0.0001**	0.382		0.633	**0.001**	**<0.0001**	0.712
		Trt	F	6.8	0.5		1.86		0.15	2.16		0.76	1.57	0.54	1.69
			df	2, 10	2, 10		2, 10		2, 10	2, 10		2, 10	2, 10	2, 10	2, 10
			P	**0.006**	0.623		0.219		0.860	0.151		0.630	0.109	0.489	0.245
		SMP*Trt	F	4.42	2.56		1.41		0.34	8.66		0.31	0.16	0.12	2.62
			df	2, 10	2, 10		2, 10		2, 10	2, 10		2, 10	2, 10	2, 10	2, 10
			P	**0.026**	0.141		0.290		0.718	**0.010**		0.422	0.856	0.982	0.127
	Site 6	SMP	F	18.45	3.88	n/a	5.71	n/a	2.34	0.03	n/a	11.59	1.01	10.21	0.73
			df	1, 10	1, 10		1, 10		1, 10	1, 10		1, 15	1, 10	1, 10	1, 10
			P	**0.001**	**0.068**		**0.042**		0.161	0.897		**0.004**	0.246	**0.012**	**0.005**
		Trt	F	2.26	2.18		2.04		0.03	0.16		0.2	0.98	0.54	3.38
			df	2, 10	2, 10		2, 10		2, 10	2, 10		2, 15	2, 10	2, 10	2, 10
			P	0.167	0.148		0.204		0.967	0.870		0.821	0.810	0.604	0.085
		SMP*Trt	F	0.86	1.01		1.68		2.01	1.58		3.51	0.66	0.95	2.14
			df	2, 10	2, 10		2, 10		2, 10	2, 10		2, 15	2, 10	2, 10	2, 10
			P	0.444	0.389		0.242		0.191	0.350		0.096	0.583	0.425	0.172
2021	Site 7	SMP	F	n/a	0.22	0.6158	7.48	n/a	0.17	1.86	n/a	1.01	9.6	38.56	22.3
			df		1, 15	1, 15	1, 15		1, 15	1, 15		1, 15	1, 15	1, 6	1, 6
			P		0.659	0.616	**0.015**		0.688	0.193		0.332	**0.029**	**0.001**	**0.005**
		Trt	F		1.27	1.2	0.79		1.72	0.21		1.15	0.33	0.56	0.44
			df		2, 15	2, 12	2, 15		2, 15	2, 15		2, 15	2, 15	2, 6	2, 6
			P		0.313	0.341	0.471		0.244	0.813		0.344	0.737	0.600	0.667
		SMP*Trt	F		1.65	0.55	0.52		2.46	0.92		0.1	0.68	0.62	0.35
			df		2, 15	2, 15	2, 15		2, 15	2, 15		2, 15	2, 15	2, 6	2, 6
			P		0.212	0.604	0.603		0.119	0.420		0.903	0.554	0.568	0.723
	Site 8	SMP	F	n/a	0.13	2.17	17.68	n/a	26.06	0.03	n/a	3.17	57.72	58.26	3.33
			df		1, 11	1, 11	1, 11		1, 11	1, 11		1, 11	1, 11	1, 11	1, 11
			P		0.815	0.099	**0.002**		**0.001**	0.882		0.326	**<0.0001**	**<0.0001**	0.101
		Trt	F		0.8	0.15	0.3		1.11	0.4		1.89	1.25	0.3	0.73
			df		2, 11	2, 11	2, 11		2, 11	2, 11		2, 11	2, 11	2, 11	2, 11
			P		0.764	0.865	0.746		0.355	0.745		0.459	0.403	0.745	0.543
		SMP*Trt	F		0.98	1.11	0.17		1.05	1.51		1.42	1.15	8.66	12.13
			df		2, 11	2, 11	2, 11		2, 11	2, 11		2, 11	2, 11	2, 11	2, 11
			P		0.697	0.451	0.843		0.382	0.499		0.511	0.367	**0.006**	**0.003**

n/a represents taxa activity lower than 1% of the total arthropod activity; therefore, analysis was not performed. SMP represents the sampling periods. Trt represents treatments. ^1^ sample period one was lost due to the rain; therefore, analysis of variance was performed for this site. ^2^ Numerator and denominator degrees of freedom, respectively. Significant *p*-values (<0.05) are shown in bold. * represent the interaction between sample (SMP) and treatment.

## Data Availability

The data presented in this study are openly available in Zenodo at: http://doi.org/10.5281/zenodo.6091653.

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
