# Peer review of "Impact of the Timing and Use of an Insecticide on Arthropods in Cover-Crop-Corn Systems"

_insects, 2022, doi:10.3390/insects13040348_

Round 1

Reviewer 1 Report

The authors seem to have made great efforts to complete this manuscript.

However, this manuscript is aimed to evaluate the use of some pesticides to control the arthropods in cover crop-corn systems to protect from several pests that can migrate from cover crop to the cash crop and their finding that the use of preventive pesticide applications is not required to protect the cash crop.

The manuscript is well written. And the description of the M&M and the results in the proper way.

Just if you can improve the introduction with more examples.

Very few comments

36 Where, in USA or worldwide?

106-107 Add coordinates.

120 Add more details about the pyrethroid insecticide (active ingredient name, concentration, production), also for glyphosate.  

129 is there some attractant inside the trap.

130 add the trap dimensions.

132 how much the plot size?

Author Response

The authors seem to have made great efforts to complete this manuscript.

However, this manuscript is aimed to evaluate the use of some pesticides to control the arthropods in cover crop-corn systems to protect from several pests that can migrate from cover crop to the cash crop and their finding that the use of preventive pesticide applications is not required to protect the cash crop.

The manuscript is well written. And the description of the M&M and the results in the proper way.

Just if you can improve the introduction with more examples.

Answer: We recently published a systematic review about how cover crop management impact arthropods (Carmona et al. 2021). In the introduction, we cited this review and highlighted the most important and relevant information to this study based on this review. If the reviewer has some specific papers, we can try to add those. Also, the journal recommendation is to keep the manuscript's main text to around 3,000 words, so we believe that more examples could be easily found with the citations we are using.

Carmona, G. I., L. M. Delserone, J. N. C. Duarte, T. F. de Almeida, D. V. B. Ozório, R., Wright, R., and A. J. McMechan. Does cover crop management affect arthropods in the subsequent corn and soybean crops in the United States? A systematic review. Ann. Entomol. Soc. Am. 2021. 2, 151-162.

Very few comments

36 Where, in USA or worldwide?

Answer: In the USA. “Projections estimate that approximately 20 million acres were planted with cover crops in 2021, with a potential increase to 100 million acres by 2025” was replaced by “In the USA, projections estimate that approximately 20 million acres were planted with cover crops in 2021, with a potential increase to 100 million acres by 2025”

106-107 Add coordinates.

Answer: We would add the coordinates if the experiments were conducted on a research station. However, as the 8-site years were conducted on growers’ private fields, we would need to get written permission from growers. We could try to obtain permission if it is mandatory for the journal Insects to have it.

120 Add more details about the pyrethroid insecticide (active ingredient name, concentration, production), also for glyphosate.  

Answer: “a.i: zeta-cypermethrin and bifenthrin was added in lines 120-121 and 123 and “Cover crop was terminated using an herbicide, Roundup PowerMAX, a.i: glyphosate (N-(phosphonomethyl) glycine3) (2.24 kg.ha-1)” was added in lines 125-126.

129 is there some attractant inside the trap.

Answer:  No attractant was used, pitfall traps were filled with RV propylene glycol-based antifreeze.  

130 add the trap dimensions.

Answer: The following was added starting on line 132. “Each pitfall trap consisted of a 473 mL cup sunken into the ground with the rim level with the soil surface. A removable 236 mL collecting cup was placed inside the larger cup, and 170 mL of propylene glycol-based antifreeze liquid with no attractant was added for each collection period to immobilize arthropods for further identification. During the pitfall collection period, a thick plastic plate was used as a cover to limit the impact of rain, at a height of 5 cm from the soil surface.”

132 how much the plot size?

Answer: All plot sizes are in Table 2 as stated in line 119 “Plot size varied between sites according to the area available (Table 2).”

Reviewer 2 Report

These are my main comments on the manuscript (Insects-1620003) entitled “Impact of the timing and use of an insecticide on arthropods in cover crop-corn systems”. The manuscript examines the insecticide effects in cover cop-systems Following moderated revisions should be incorporated in the manuscript prior to acceptance.
Ls.11-12: Revise this sentence to eliminate rewordiness.
L.32: Keywords should be in alphabetic order. Also, keywords serve to widen the opportunity to be retrieved from a database. To put words that already are into title and abstracts makes KW not useful. Please choose terms that are neither in the title nor in abstract.
L.59: Delete “noctuid”
Ls.64-66: Revise this sentence to eliminate rewordiness.
Ls.78-82: In this manuscript, insert brief information about mode of action of pyrethroid and herbicide effects on insects.
Ls.97-101: One only hypothesis is needed. Combine and summarize these sentences.
L.111: More information (chemical molecule, manufacturers, formulation, etc.) of pyrethroid insecticide is needed. 
Ls.298-331: Figures 1-6 should be organized in one or two plates.

Author Response

These are my main comments on the manuscript (Insects-1620003) entitled “Impact of the timing and use of an insecticide on arthropods in cover crop-corn systems”. The manuscript examines the insecticide effects in cover cop-systems Following moderated revisions should be incorporated in the manuscript prior to acceptance.

Ls.11-12: Revise this sentence to eliminate rewordiness.

Answer: “As a strategy to minimize pest transitions, growers may apply insecticides at the cover crop termination time as a preventive strategy” was replaced for “As a preventative strategy to minimize pest transitions, growers may apply insecticides at the cover crop termination time”.

L.32: Keywords should be in alphabetic order. Also, keywords serve to widen the opportunity to be retrieved from a database. To put words that already are into title and abstracts makes KW not useful. Please choose terms that are neither in the title nor in abstract.

Answer: We checked other articles published with Insects, and we noticed that the keywords are used to link to other similar articles in the journal. With that, we believe that it would be better to keep the keywords we are using because those describe very well the most important topics of our manuscript. We reorganized the keywords to be in alphabetic order.

L.59: Delete “noctuid” –

Answer: “noctuid” was deleted.

Ls.64-66: Revise this sentence to eliminate rewordiness.

Answer: “The transition time of the cover crop pest to the cash crop is not expected to occur immediately after cover crop termination but instead when the cover crop is deteriorating” was replaced for “The transition time of the cover crop pest to the cash crop is expected to occur when the cover crop is deteriorating”

Ls.78-82: In this manuscript, insert brief information about mode of action of pyrethroid and herbicide effects on insects.

Answer: Starting in line 85, “Insecticides are used due to their fast action, easy application, and potential economic return.” Was replaced for “Pyrethroid insecticides are often used due to their fast action, easy application, low cost, and potential economic return.” In lines 85-86. The following sentences describe the impact of pyrethroid on insects.

L.111: More information (chemical molecule, manufacturers, formulation, etc.) of pyrethroid insecticide is needed. 

Answer: “a.i: zeta-cypermethrin and bifenthrin was added in lines 120-121 and 123 and “Cover crop was terminated using an herbicide, Roundup PowerMAX, glyphosate (N-(phosphonomethyl) glycine3) (2.24 kg.ha-1)” was added in lines 125-126.

Ls.97-101: One only hypothesis is needed. Combine and summarize these sentences.

Answer: “We hypothesized that the addition of any preventive insecticide application would negatively impact arthropod activity. We also hypothesize that the addition of an insecticide application would not increase corn yields in a cover crop to corn systems unless significant pest pressure occurred.” was replaced for “We hypothesized that the addition of any preventive insecticide application would reduce arthropod activity and would not increase corn yields unless significant pest pressure occurred.”

Ls.298-331: Figures 1-6 should be organized in one or two plates.

Answer: Thanks for this suggestion. We made a panel of Figs 1,2,3,4,5,6, having now Figure 1A, B, C, D, E, F.

Reviewer 3 Report

The authors present an interesting study of insecticide management on arthropods in a cover crop-cash crop rotation to help guide better management practices. Extensive study with impressive data collection and sorting to relevant group but the ms could be improved by more detailed analysis of impacts on groups of particular interest.

I fully support the objective of this manuscript to investigate the impact of insecticide management on arthropods in cover crop-cash crop rotation systems. Inclusion of herbicides especially of interest. Understanding of the impacts of chemicals on both pests and natural enemies is important to supporting choices that maximize the contribution of natural enemies to pest control while not enhancing pest abundance. This is an important and challenging research area. However, the outcome could be improved by finer scale analysis of the arthropods collected. The impressive data set presented has potential to be more informative than as presented here.

Analysis of total arthropods, given that this potentially includes both pest and natural enemy, is of limited use. The analysis would be improved if impacts on identified taxa of interest could be presented and then advice as to outcomes of management plans determined.

Conclusions could be clarified to give more definite information about a range of organisms of interest. Too much emphasis on carabids and dead matter.

Here are my specific points:

L100 both pest and beneficial?

P values more usually to 3 dp and <0.001 lowest

Were there impacts on pest numbers in the cash crop?

Author Response

The authors present an interesting study of insecticide management on arthropods in a cover crop-cash crop rotation to help guide better management practices. Extensive study with impressive data collection and sorting to relevant group but the ms could be improved by more detailed analysis of impacts on groups of particular interest.

 I fully support the objective of this manuscript to investigate the impact of insecticide management on arthropods in cover crop-cash crop rotation systems. Inclusion of herbicides especially of interest. Understanding of the impacts of chemicals on both pests and natural enemies is important to supporting choices that maximize the contribution of natural enemies to pest control while not enhancing pest abundance. This is an important and challenging research area. However, the outcome could be improved by finer scale analysis of the arthropods collected. The impressive data set presented has potential to be more informative than as presented here.

 Analysis of total arthropods, given that this potentially includes both pest and natural enemy, is of limited use. The analysis would be improved if impacts on identified taxa of interest could be presented and then advice as to outcomes of management plans determined.

 Answer: We collected a total of 33,316 arthropods during the three sampling periods using pitfall traps. We agree that focusing only on the total arthropod would not provide much information. However, we analyzed the total arthropod and the most abundant individual taxa, that is, the taxa that corresponded to at least 1% of the total arthropod activity for each site and year (see lines 167-172). You can find the P-values (all fixed effects and interactions) for all arthropod taxa that was analyzed individually and for the total arthropods in table 4. We framed our discussion around the taxa that were influenced by the study treatments. For the analysis strategy, we followed the procedure done by Dunbar et al. (2016).

As you can see in Table 4, most of the arthropod taxa were not influenced by any insecticide application (shown in Table 4 as “treatment”). These results led us to our hypothesis that the cover crop biomass might be sheltering the soil-dwelling arthropods from the insecticide applications. Based on our findings regarding the tank mix application decreasing Aphididae and increasing Carabidae, we hypothesized that 1) Increase in Carabids with the tank-mix application was led by impaired or dead Aphids in the same site and treatment, and 2) potential insecticide hormesis effect on Carabids, characterized by a low-dose response that is opposite in effect to that seen at high doses.

Dunbar M. W., A. J. Gassmann, and M. E. O’Neal. Limited impact of fall-Seeded, spring-terminated rye cover crop on beneficial arthropods. J. Environ. Entomol. 2017. 46(2), 284-290.

References supporting hypothesis regarding Carabidae and Aphididae prey-predator interaction with insecticides application:

  1. Chiverton P. A. Pitfall-trap catches of the carabid beetle Pterostichus melanarius, in relation to gut contents and prey densities, in insecticide treated and untreated spring barley.  Exp. Appl. 1984. 36,23-30.
  2. Heneghan P. A. Assessing the effects of an insecticide on the activity of predatory ground beetles, in Interpretation of Pesticide Effects on Beneficial Arthropods. of Appl. Biologists1992. Wellesbourne, UK, pp. 113-119.
  3. Bel'skaya E. A., E.V. Zinov'ev, and M.A. Kozyrev. Carabids in a spring wheat agrocenosis to the south of Sverdlovsk oblast and the effect of insecticide treatment on their populations.  J. Ecol.2002. 33, 38– 44.

References supporting hypothesis regarding hormesis effect on Carabidae:

  1. Calabrese, E. J. Paradigm lost, paradigm found: the re-emergence of hormesis as a fundamental dose response model in the toxicological sciences.  Pollut.2005. 138, 379–411.
  2. Cohen, E. Pesticide-mediated hemeostatic modulation in arthropods.  Biochem. Physiol.2006. 85, 21–27.
  3. Cutler, G. C. Insects, insecticides and hormesis: Evidence and considerations for study. Dose Response. 11, 154–177.

Conclusions could be clarified to give more definite information about a range of organisms of interest. Too much emphasis on carabids and dead matter.

Answer: A conclusion section was added to the manuscript starting in line 449. The comment above provides an explanation about how we conducted the analysis and why we are focusing on some specific arthropod taxa.

 Here are my specific points:

L100 both pest and beneficial?

Answer: Yes. We hypothesized that the use of any preventive insecticide application would reduce the arthropod activity, pest or beneficial. Moreover, we hypothesized that the insecticide applications would not reduce corn grain yield unless significant pest pressure occurred. Based on the other reviewer’s suggestion, we rewrote the hypothesis: “We hypothesized that the addition of any preventive insecticide application would negatively impact arthropod activity. We also hypothesize that the addition of an insecticide application would not increase corn yields in a cover crop to corn systems unless significant pest pressure occurred.”. This was replaced for “We hypothesized that the addition of any preventive insecticide application would reduce arthropod activity and would not increase corn yields unless significant pest pressure occurred.”

P values more usually to 3 dp and <0.001 lowest:

Answer: All p-values in Table 4 and throughout the results were edited to have 3 dp.

Were there impacts on pest numbers in the cash crop?

Answer: No. Corn Injury Assessments were done at V3 corn stage and less than 1% of pest pressure was found; therefore, data were not analyzed due to the low numbers and lack of significance. “Future research should consider artificial pest infestation to further evaluate the impact of insecticide use in cover crop systems and corn grain yield trades-off.” Was added in lines 438-439 to guide future research in this topic.

Round 2

Reviewer 3 Report

Authors have responded adequately to earlier review. I find the ms suitable for publication